# Tirzepatide: A Systematic Update

**DOI:** 10.3390/ijms232314631

**Published:** 2022-11-23

**Authors:** Imma Forzano, Fahimeh Varzideh, Roberta Avvisato, Stanislovas S. Jankauskas, Pasquale Mone, Gaetano Santulli

**Affiliations:** 1Department of Advanced Biomedical Sciences, Division of Cardiology, “*Federico II*” University, 80131 Naples, Italy; 2Department of Medicine, Division of Cardiology, Wilf Family Cardiovascular Research Institute, Einstein Institute for Aging Research, Albert Einstein College of Medicine, New York, NY 10461, USA; 3Department of Molecular Pharmacology, Einstein Institute for Neuroimmunology and Inflammation (*INI*), Einstein-Mount Sinai Diabetes Research Center (*ES-DRC*), Fleischer Institute for Diabetes and Metabolism (*FIDAM*), Albert Einstein College of Medicine, New York, NY 10461, USA

**Keywords:** cardiovascular medicine, diabetes, hypertension, GIP, GLP-1, glucagon, incretins, LY3298176, meta-analysis, obesity, SUMMIT, SURMOUNT, SURPASS, SYNERGY, Tirzepatide

## Abstract

Tirzepatide is a new molecule capable of controlling glucose blood levels by combining the dual agonism of Glucose-Dependent Insulinotropic Polypeptide (GIP) and Glucagon-Like Peptide-1 (GLP-1) receptors. GIP and GLP1 are incretin hormones: they are released in the intestine in response to nutrient intake and stimulate pancreatic beta cell activity secreting insulin. GIP and GLP1 also have other metabolic functions. GLP1, in particular, reduces food intake and delays gastric emptying. Moreover, Tirzepatide has been shown to improve blood pressure and to reduce Low-Density Lipoprotein (LDL) cholesterol and triglycerides. Tirzepatide efficacy and safety were assessed in a phase III SURPASS 1–5 clinical trial program. Recently, the Food and Drug Administration approved Tirzepatide subcutaneous injections as monotherapy or combination therapy, with diet and physical exercise, to achieve better glycemic blood levels in patients with diabetes. Other clinical trials are currently underway to evaluate its use in other diseases. The scientific interest toward this novel, first-in-class medication is rapidly increasing. In this comprehensive and systematic review, we summarize the main results of the clinical trials investigating Tirzepatide and the currently available meta-analyses, emphasizing novel insights into its adoption in clinical practice for diabetes and its future potential applications in cardiovascular medicine.

## 1. Introduction

Type 2 Diabetes Mellitus (T2DM) is an important independent risk factor for atherosclerotic cardiovascular disease [1], is frequently associated with obesity, and has been linked to high morbidity and mortality rates [2]. Glucose-Dependent Insulinotropic Polypeptide (GIP) and Glucagon-Like Peptide-1 (GLP-1), also known as incretins, are hormones released in the intestine in response to the intake of nutrients, and they are capable of stimulating pancreatic beta cells to release insulin, thereby participating in the regulation of glucose homeostasis [3,4,5].

GIP inhibits gastric secretion activity, stimulates insulin secretion, and has insulin-like action on adipose tissue inhibiting lipolysis and promoting lipogenesis [6,7,8]. GLP1 is able to stimulate insulin secretion and inhibit glucagon release; it also slows gastric emptying and induces a sense of satiety [9]. Incretins are rapidly degraded by dipeptidyl-peptidase 4 (DPP4) [10,11]. Thus, DPP4 inhibitors and GLP1 receptor agonists have been used to date as drugs for the treatment of T2DM acting on the incretin system [12,13,14]. Mounting evidence indicates that the contemporary administration of GIP and GLP1 has a synergistic effect by significantly increasing insulin response and glucagonostatic action [15,16,17,18,19,20,21,22,23,24,25].

A new molecule with a combined agonist action on both GPI and GLP1 receptors (“twincretin”) has been developed to take advantage of this synergistic effect: Tirzepatide (LY3298176) [26,27,28] is the first “twincretin”, a synthetic peptide composed of 39 amino acids based on the GIP native sequence [29,30], combining the dual agonism of GIP and GLP-1 receptors controlling glycemic blood level and reducing body weight (Figure 1). Tirzepatide has an affinity for the GIP receptor equal to that of native GIP and an affinity for the GLP-1 receptor ~5-fold weaker than that of native GLP-1 [31,32]. Tirzepatide can also improve parameters related to cardiovascular risk, including blood pressure (BP) [33], waist circumference, LDL, and circulating triglycerides [34,35,36,37]. Both its effectiveness and safety have been demonstrated in several clinical trials.

In this comprehensive review, we analyze the main results of the trials investigating Tirzepatide and discuss its future use as a drug to tackle T2DM and cardiovascular disorders.

## 2. Tirzepatide in the SURPASS Trials

The *Study of Tirzepatide in Participants with T2DM Not Controlled with Diet and Exercise Alone* (SURPASS) clinical trials were designed to demonstrate the effectiveness and safety of Tirzepatide as a hypoglycemic drug in patients affected by T2DM. The primary outcome was the mean change in glycated hemoglobin (HbA1c) from baseline.

SURPASS-1 was a randomized, double blinded, clinical trial that evaluated the effectiveness of Tirzepatide (subcutaneous injections once weekly) compared to placebo in patients with T2DM inadequately controlled by diet and exercise [38]. This trial demonstrated that Tirzepatide is superior to placebo in improving glycemic control and reducing body weight [38]. Indeed, Tirzepatide, at all doses tested, was better than placebo in inducing changes from baseline in HbA1c, fasting serum glucose, body weight, and HbA1c targets of less than 7.0% (<53 mmol/mol) and less than 5.7% (<39 mmol/mol) at 40 weeks. Participants were randomly assigned to Tirzepatide 5 mg (n = 121), Tirzepatide 10 mg (n = 121), Tirzepatide 15 mg (n = 120), or placebo (n = 113) (Table 1).

In summary, SURPASS-1 established the effectiveness of treatment with Tirzepatide once a week at doses of 5, 10, and 15 mg as monotherapy for T2DM compared with placebo in improving glycemic control. Tirzepatide exhibited robust effectiveness compared with placebo in glycemic control, with 31–52% of participants reaching normoglycemia (HbA1c < 5.7%; <39 mmol/mol) and meaningful reductions in body weight without an increased risk of clinically significant or severe hypoglycemia (<54 mg/dL; <3 mmol/L). The range of weight reduction was from 7 to 9.5 Kg. The safety profile was consistent with GLP-1 receptor agonists [39,40].

SURPASS-2 was an open-label, phase III trial that compared Tirzepatide with the GLP-1R agonist Semaglutide once weekly in patients with T2DM [41]. Tirzepatide at all doses was noninferior and superior to Semaglutide with respect to the mean changes in HbA1c levels from baseline at 40 weeks [41]. Patients were randomly assigned in a 1:1:1:1 ratio to receive Tirzepatide at a dose of 5 mg (n = 470), 10 mg (n = 469), or 15 mg (n = 470) or Semaglutide (n = 469) at a dose of 1 mg. The mean changes from baseline in HbA1c levels were −2.01% with Tirzepatide 5 mg, −2.24% with Tirzepatide 10 mg, −2.30 % with Tirzepatide 15 mg, and −1.86% with Semaglutide; the differences between the 5 mg, 10 mg, and 15 mg Tirzepatide groups and the Semaglutide group were −0.15 %, −0.39 %, and −0.45%, respectively.

The HbA1c level target of less than 5.7% (normoglycemia) was met in 27 to 46% of the patients who received Tirzepatide and in 19% of the patients who received Semaglutide. Reductions in body weight were greater with Tirzepatide (from −7.6 to −11.2 Kg) than with Semaglutide (−5.7 Kg); the differences were −1.9 kg, −3.6 kg, and −5.5 kg with Tirzepatide 5 mg, 10 mg, and 15 mg, respectively (Table 2). These findings suggest that an increased prescription of Tirzepatide could have favorable effects on the control of diabetes and obesity [42].

An adjusted indirect treatment comparison, using aggregate data from the SURPASS-2 study that met the HbA1c inclusion criterion of the *Research Study to Compare Two Doses of Semaglutide Taken Once Weekly in People with Type 2 Diabetes* (SUSTAIN FORTE trial) and from SUSTAIN FORTE metformin-only-treated patients, confirmed that HbA1c and weight reductions were significantly greater for Tirzepatide 10 and 15 mg versus Semaglutide 2 mg [43].

In the SURPASS-3 trial, once-weekly Tirzepatide was compared with once-daily insulin degludec as an add-on to metformin with or without inhibitors of Sodium/Glucose Transporter 2 (SGLT2) in patients with T2DM not adequately controlled [44]. In this open-label, phase 3 study, participants were randomly assigned (1:1:1:1) to once-weekly subcutaneous injections of Tirzepatide 5 (n = 358), 10 (n = 360), or 15 (n = 358) mg or once-daily subcutaneous injections of titrated insulin degludec (n = 359). The mean changes from baseline in the HbA1c levels were −1.93% with Tirzepatide 5 mg, −2.20% with Tirzepatide 10 mg, −2.37% with Tirzepatide 15 mg, and −1.34% with insulin degludec. The differences between the 5 mg, 10 mg, and 15 mg Tirzepatide groups and the degludec group were −0.59%, −0.86%, and −1.04%, respectively (Table 3). In this study, up to 93% of participants receiving Tirzepatide achieved the HbA1c target of less than 7.0%, and 26–48% of participants treated with Tirzepatide achieved normoglycemia (HbA1c < 5.7%; <39 mmol/mol) [45].

Reductions in body weight were greater with Tirzepatide than with insulin degludec. The differences were −9.8 kg with Tirzepatide 5 mg, −13 kg with Tirzepatide 10 mg, and −15.2 kg with Tirzepatide 15 mg. Once-weekly treatment with Tirzepatide also led to superior glycemic control, measured using continuous glucose monitoring (CGM), compared to insulin degludec in participants with T2DM on metformin, with or without an SGLT2 inhibitor [46]. A sub-study of the SURPASS-3 trial [47], using magnetic resonance imaging (MRI), demonstrated that Tirzepatide was also able to significantly reduce liver fat content (LFC) [48], the volume of visceral adipose tissue (VAT), and abdominal subcutaneous adipose tissue (ASAT) [47].

In order to provide an integrated measure of the physiological effects of Tirzepatide on glucose homoeostasis, a double-blind, randomized, parallel-arm, phase 1 study that compared the hormonal and metabolic effects of Tirzepatide (titrated to once-weekly 15 mg) with those of Semaglutide (titrated to once-weekly 1 mg) or placebo in patients with T2DM was designed, using gold-standard dynamic stimulatory tests to assess insulin secretion and insulin sensitivity (i.e., hyperglycemic and hyperinsulinemic euglycemic clamps) and using mixed-meal tolerance tests [49]. The trial revealed that the increase in the clamp disposition index, which adjusts insulin secretion for concurrent insulin sensitivity, was significantly larger in the Tirzepatide arm compared to the Semaglutide arm (*p* < 0.0001), with improvements in both the total insulin secretion rate and insulin sensitivity [49], the latter possibly being related to a greater weight loss (–11.2 kg vs. –6.9 kg) [50]. The responses to the mixed-meal tolerance test indicated reduced glucose excursions with Tirzepatide compared to Semaglutide, associated with lower insulin and glucagon plasma concentrations, strongly suggesting a diminished insulin resistance and a reduced workload for pancreatic β cells [49,50,51].

In the SURPASS-4 trial, Tirzepatide was compared to insulin glargine in patients with a high cardiovascular risk, suffering from T2DM inadequately controlled by oral glucose-lowering medications [52]. Patients were randomly assigned to receive at least one dose of Tirzepatide 5 mg (n = 326), 10 mg (n = 321), 15 mg (n = 334), or insulin glargine (n = 978). The study demonstrated that all three doses of Tirzepatide markedly improved glucose control, reduced body weight, and improved the cardiovascular risk profile in these patients. Indeed, at 52 weeks, the mean HbA1c changes with Tirzepatide were −2.24% with 5 mg, –2.43% with 10 mg, and –2.58% with 15 mg versus –1.44% with insulin glargine [53]. The treatment difference versus insulin glargine was −0.80% for Tirzepatide 5 mg, −0.99% for Tirzepatide 10 mg, and −1.14% for Tirzepatide 15 mg [54]. The percentage of patients achieving normoglycemia using Tirzepatide was 23–43%. The weight loss from baseline was −7.1 Kg, −9.5 Kg, and −11.7 Kg for patients treated with Tirzepatide 5 mg, 10 mg, and 15 mg, respectively, and the weight gain from baseline was +1.9 Kg in patients treated with insulin glargine. The treatment differences versus insulin glargine were −9 Kg for Tirzepatide 5 mg, −11.4 Kg for Tirzepatide 10 mg, and −13.5 Kg for Tirzepatide 15 mg (Table 4).

Important limitations of the SURPASS-4 trial should be noted, including the fact that the interventions were not blinded because of differences in devices or dose escalation schemes and that not all participants were treated for 104 weeks; moreover, the study only included patients with a body mass index (BMI) ≥ 25 kg/m^2^ and a stable weight (≤5% fluctuation in either direction) during the previous 3 months; thus, the effects in subjects with a lower BMI remain to be determined. Intriguingly, a post hoc analysis of the SURPASS-4 trial demonstrated that, in people with T2DM and a high cardiovascular risk, Tirzepatide slowed the rate of the decline of the estimated glomerular filtration rate (eGFR) and reduced the urine albumin–creatinine ratio in clinically meaningful ways compared with insulin glargine [55].

In the SURPASS-5 phase III trial, among patients with T2DM and inadequate glycemic control despite treatment with insulin glargine, the addition of subcutaneous Tirzepatide, compared with placebo, to titrated insulin glargine resulted in statistically significant improvements in glycemic control after 40 weeks [56]. Patients were randomized in a 1:1:1:1 ratio to receive once-weekly subcutaneous injections of 5 mg (n = 116), 10 mg (n = 119), or 15 mg (n = 120) Tirzepatide or volume-matched placebo (n = 120) over 40 weeks. At the end of the study, the mean HbA1c changes from baseline were −2.11% with 5 mg Tirzepatide, −2.40% with 10 mg Tirzepatide, and −2.34% with 15 mg Tirzepatide versus −0.86% with placebo. The differences between the 5 mg, 10 mg, and 15 mg Tirzepatide groups and the placebo group were −1.24%, −1.53%, and −1.47%, respectively. The proportion of patients treated with Tirzepatide, at all doses, reaching normoglycemia was 24.4–49.6%. The mean body weight changes from baseline were −5.4 kg with 5 mg Tirzepatide, −7.5 kg with 10 mg Tirzepatide, −8.8 kg with 15 mg Tirzepatide, and +1.6 kg with placebo [57]. The differences between the 5 mg, 10 mg, and 15 mg Tirzepatide groups and the placebo group were −7.1 kg, −9.1 kg, and −10.5 kg, respectively (Table 5).

In SURPASS J-mono, a randomized phase III clinical trial conducted on Japanese patients affected by T2DM, Tirzepatide at doses of 5 mg, 10 mg, and 15 mg was compared to the GLP-1 agonist Dulaglutide (0.75 mg). Tirzepatide was superior to Dulaglutide for both glycemic control and reductions in body weight. The safety profile of Tirzepatide was consistent with that of GLP-1 receptor agonists, indicating a potential therapeutic use in Japanese patients with T2DM [58]. Compared to Dulaglutide, Tirzepatide showed greater potential for normalizing metabolic factors after a standardized meal, significantly reducing body weight and body fat mass [59].

Similarly, the SURPASS J-combo assessed the safety and glycemic efficacy of Tirzepatide as an add-on treatment in Japanese patients with T2DM who had inadequate glycemic control with stable doses of various oral antihyperglycemic monotherapies. This trial confirmed that Tirzepatide is well-tolerated as an add-on to oral antihyperglycemic monotherapy in Japanese participants with T2DM and demonstrated a significant improvement in glycemic control and reductions in body weight, irrespective of the background oral antihyperglycemic medication [60].

## 3. Tirzepatide and Obesity: The SURMOUNT-1 Trial

Obesity and, consequently, excess adiposity are related to numerous complications, including hypertension, dyslipidemia, and T2DM. In particular, body mass index (BMI) and waist circumference are known to be associated with T2DM and cardiovascular disease [61,62,63,64]. Obesity contributes to the development of cardiovascular disease and to cardiovascular mortality independently of other cardiovascular risk factors [65]. In the SURPASS 1–5 trials, weight reduction after Tirzepatide treatment was significant; BMI and waist circumference were analyzed showing a significant reduction in these parameters, suggesting a better metabolic profile in response to Tirzepatide. These results are very encouraging for the use of Tirzepatide to treat patients affected by obesity with or without T2DM.

The *Study of Tirzepatide in Participants with Obesity or Overweight* (SURMOUNT-1), a phase III, double-blind, randomized, controlled clinical trial, revealed that, in participants with a confirmed diagnosis of obesity, Tirzepatide 5 mg, 10 mg, or 15 mg once a week for 72 weeks provided substantial and sustained reductions in body weight [66]. Tirzepatide improved cardiometabolic measures, such as waist circumference. The exclusion criteria were diabetes, a change in body weight of more than 5 kg within 90 days before screening, previous or planned surgical treatment for obesity, and treatment with a medication that promotes weight loss within 90 days before screening. The co-primary endpoints were the percentage change in weight from baseline and a weight reduction of 5% or more. Participants were randomly assigned to receive Tirzepatide administered subcutaneously in a 1:1:1:1 ratio at a dose of 5 mg (n = 630), 10 mg (n = 636), or 15 mg (n = 636) or placebo (n = 643) once a week for 72 weeks as an adjunct to lifestyle intervention. To obtain a meaningful effect on the basis of improvement in metabolic health, there should be a body weight reduction of 5% or more [67]. In the SURMOUNT-1 trial, the mean percentage changes in weight at the end of the study were 15.0%, −19.5%, and −20.9% with Tirzepatide 5 mg, 10 mg, and 15 mg, respectively, and −3.1% with placebo. The percentages of participants who had a weight reduction of 5% or more were 85%, 89%, and 91% with Tirzepatide 5 mg, 10 mg, and 15 mg, respectively, and 35% with placebo; 50% of participants in the 10 mg group and 57% of participants in the 15 mg group had a reduction in body weight of 20% or more, as compared with 3% in the placebo group [68,69,70]. Reductions in waist circumferences were also observed: in the Tirzepatide group, there was a mean reduction in waist circumference of −14 cm with 5 mg, −17.7 with 10 mg, and 18.5 cm with 15 mg versus a reduction of 4 cm with placebo. The estimated differences between the 5 mg, 10 mg, and 15 mg Tirzepatide groups and the placebo group were −10.1 cm, −13.8 cm, and −14.5 cm, respectively (Table 6).

Of course, further studies are necessary to fully establish the effectiveness of Tirzepatide in improving lipid profile and metabolic syndrome and in reducing cardiovascular risk independently of glycemic status. Ideally, future studies should have a longer follow-up period. Indeed, Benfluorex, a fenfluramine derivative designed for weight loss, was associated with mitral regurgitation after two years of use in a case–control retrospective study [71].

## 4. Tirzepatide and Hypertension

Although hypertension is known to be associated with an increased risk of cardiovascular disease and death, BP control remains suboptimal [72,73]. Tirzepatide has been shown to favorably affect BP [33], and this result has been consistently confirmed in all trials. Indeed, in the SURPASS-1 trial, the mean systolic BP (SBP) decrease ranged from −4.7 to −5.2 mmHg versus −2.00 mmHg with placebo, whereas diastolic BP (DBP) did not significantly differ with placebo; in the SURPASS-2 trial, SBP and DBP decreased to −4.8 mmHg and −1.9 mmHg, respectively, with Tirzepatide at a dose of 5 mg; −5.3 mm Hg and −2.5 mmHg, respectively, at a dose of 10 mg; and −6.5 mm Hg and −2.9 mm Hg, respectively, at a dose of 15 mg; versus −3.6 mmHg and −1.0 mm Hg, respectively, with Semaglutide. In the SURPASS-3 trial, significant decreases in mean SBP of −4.9 to −6.6 mm Hg and in mean DBP of −1.9 to −2.5 mm Hg were observed for Tirzepatide, while no significant changes were detected for insulin degludec. In the SURPASS-4 trial, mean SBP and DBP decreased with Tirzepatide from −2.8 to −4.8 mmHg and from −0.80 to −1.0 mmHg, respectively, and increased with insulin glargine: a +1.3 mm Hg increase in SBP and a +0.7 mmHg increase in DBP. In the SURPASS-5 trial, the mean changes in SBP and DBP were −6.1 to −12.6 mm Hg and −2.0 to −4.5 mmHg, respectively, for the Tirzepatide groups and −1.7 mm Hg and −2.1 mm Hg for the placebo group. Similarly, in the SURMOUNT-1 trial, participants treated with Tirzepatide reported a mean reduction in SBP of −7.2 mmHg and a mean reduction in DBP of −4.8 mmHg. In the placebo group, a mean reduction in SBP of −1 mmHg and a reduction in DBP of −0.8 mmHg were observed (Figure 2).

The beneficial effects of Tirzepatide on BP could be attributable to an amelioration of endothelial function and/or to reduced inflammation [33], although dedicated studies are needed to define the exact mechanisms underlying these findings. Nevertheless, a post hoc analysis from the phase IIb trial showed that Tirzepatide was associated with a dose-dependent decrease from baseline in the levels of high-sensitivity C-reactive protein, intercellular adhesion molecule 1 (ICAM-1), and YKL-40 at 26 weeks [26,74,75].

## 5. Data from Meta-Analyses

Different meta-analyses have been conducted to analyze the effectiveness and safety of Tirzepatide. Sattar and collaborators, scrutinizing all studies of the SURPASS program, compared the time to first occurrence of the well-established four major adverse cardiovascular events (MACE-4; cardiovascular death, myocardial infarction, stroke, and hospitalized unstable angina) between pooled Tirzepatide groups and control groups [76]. The stratified Cox proportional hazards model was used (fixed effect: treatment; stratification factor: trial-level cardiovascular risk) for the estimation of hazard ratios (HRs) and confidence intervals (CIs) comparing Tirzepatide to controls. They underlined that, despite the favorable effects of Tirzepatide on a range of CV risk factors, only the results from the SURPASS-4 trial have hitherto reported data on the CV safety of the drug. After their analysis, they concluded that Tirzepatide does not increase the risk of MACE-4 in participants with T2DM (HRs comparing Tirzepatide versus controls were as follows: for MACE-4, 0.80, 95% CI, 0.57–1.11; for cardiovascular death, 0.90, 95% CI, 0.50–1.61; and for all-cause death, 0.80, 95% CI, 0.51–1.25), even if the exclusion of people with unstable cardiovascular disease (such as class IV heart failure) is a limitation [76].

Karagiannis and co-workers [77] conducted a meta-analysis to assess the efficacy and safety of Tirzepatide in T2DM. The results with Tirzepatide demonstrated a dose-dependent superiority on glycemic efficacy and body weight reduction compared to placebo, GLP-1 receptor agonists, and basal insulin. Tirzepatide was associated with an increased incidence of gastrointestinal adverse events: nausea was the most frequent event with all Tirzepatide doses, especially 15 mg (OR 5.60, 95% CI 3.12 to 10.06); Tirzepatide 15 mg was associated with higher incidences of vomiting (OR 5.50, 95% CI 2.40 to 12.59) and diarrhea (OR 3.31, 95% CI 1.40 to 7.85). The risk of hypoglycemia did not increase with Tirzepatide. The presence of statistical heterogeneity in the meta-analyses for changes in HbA1c and body weight, the assessment of the risk of bias solely for the primary outcome, and the generalization of findings mainly to individuals with overweight or obesity and already on metformin-based background therapy were acknowledged as study limitations [77]. Another pooled analysis [78] confirmed that Tirzepatide treatment resulted in a greater lowering of HbA1c (−1.94%, 95% CI: −2.02 to −1.87), fasting serum glucose (−54.72 mg/dL, 95% CI: −62.05 to −47.39), and body weight (−8.47, 95% CI: −9.66 to −7.27); as far as the safety profile is concerned, the results of this meta-analysis were essentially consistent with those of the above-mentioned analysis conducted by Karagiannis et al. [77]. Similarly, Dutta and colleagues [79] found that individuals receiving Tirzepatide for over 1 year had a significantly greater lowering of HbA1c (−0.75%, 95% CI: −1.05 to −0.45; *p* <0.01), fasting glucose (−0.75 mmol/L, 95% CI: −1.05 to −0.45; *p* < 0.01), 2 h post-prandial-glucose (−0.87 mmol/L, 95% CI: −1.12 to −0.61; *p* < 0.01), weight (−8.63 kg, 95% CI: −12.89 to −4.36; *p* < 0.01), body mass index (−1.80 kg/m^2^, 95% CI: −2.39 to −1.21; *p* < 0.01), and waist circumference (−4.43 cm, 95% CI: −5.31 to −3.55; *p* < 0.01) than individuals receiving Dulaglutide, Semaglutide, insulin degludec, or glargine. According to Guan and collaborators [80] Tirzepatide 10 and 15 mg had better antidiabetic and weight-loss effects (especially the 15 mg dose) compared to insulin (glargine or degludec) and selective GLP1 receptor agonists (Dulaglutide or Semaglutide once a week); Tirzepatide 15 mg greatly reduced glycated hemoglobin (surface under the cumulative ranking curve value, SCURA probability: 93.5%), body weight (99.7%), and fasting serum glucose (86.6%). Insulin caused less gastrointestinal events (93.5%), and there was no statistical difference between GLP1-RA and Tirzepatide. They concluded that additional well-designed RCTs are needed to evaluate its clinical performance with higher doses of GLP1 receptor agonists and to definitively determine the potential cardiovascular benefits [80]. In an evaluation of the optimal dose of Tirzepatide for the treatment of T2DM using a meta-analysis and a trial sequential analysis (TSA), Tirzepatide 15 mg was superior to 10 mg and 5 mg for lowering glycemia and reducing weight; Tirzepatide 5 mg was superior to 10 mg and 15 mg (which appear to have the same effect of the 10 mg) in terms of safety [81].

In their meta-analysis of randomized clinical trials on the efficacy and safety of Tirzepatide as a novel treatment for T2DM, Permana and colleagues concluded that this drug has shown superiority in glycemic control and body weight reduction with a good safety profile in patients with T2DM [82]. Lisco et al. considered eleven clinical trials and concluded that Tirzepatide provides a weight loss that exceeds that obtained with GLP-1 receptor agonists; hence, Tirzepatide is presented as a potent tool to improve glucose control without increasing hypoglycemic risk in poorly controlled T2DM treated with basal insulin with or without other hypoglycemic oral agents with effects on body weight loss, despite the background therapy [83].

Exploring the impact of Tirzepatide on cardiovascular disorders, Patoulias and co-workers conducted a meta-analysis assessing the effect of Tirzepatide on the risk of atrial fibrillation in patients with T2DM. Pooling data from SURPASS-2 to -5, they demonstrated that Tirzepatide compared with placebo or an active comparator did not have a significantly different effect on the risk of atrial fibrillation (risk ratio = 1.59; 95% CI: 0.46 to 5.47; *p* = 0.47) [84]. These findings remain to be confirmed in the forthcoming SURPASS-CVOT, a large phase 3, randomized, double-blind, cardiovascular outcome trial, assessing both the noninferiority and superiority of Tirzepatide against Dulaglutide. Nevertheless, in a subsequent study analyzing eight trials, the same group evidenced that Tirzepatide resulted in a significantly reduced risk of major averse cardiovascular events by 48% compared to a control (RR 0.52, 95% CI 0.38 to 0.72); moreover, Tirzepatide displayed a significantly attenuated risk of cardiovascular death (by 49%; RR 0.51, 95% CI 0.29 to 0.89), as well as all-cause death (by 49%; RR 0.51, 95% CI 0.34 to 0.77) [85].

In conclusion, according to the data currently available, it is reasonable to speculate that, despite the warranted need of other randomized clinical trials, Tirzepatide may be considered a new powerful tool to treat patients with diabetes and/or obesity.

## 6. New Perspectives

The effectiveness and safety profile of Tirzepatide have opened a wide range of perspectives for its use not limited to patients with T2DM and patients with obesity [86,87].

For instance, in *post hoc* analyses, the effects of Tirzepatide on biomarkers of non-alcoholic fatty liver disease (NAFLD) and fibrosis in patients with T2DM compared to Dulaglutide and placebo for 26 weeks have also been investigated, showing that a higher Tirzepatide dose significantly decreases NAFLD-related biomarkers and increases adiponectin in patients with T2DM [88].

Metabolomics and lipidomics analyses conducted in >250 patients revealed that a 26-week treatment with Tirzepatide significantly modulated a cluster of metabolites and lipids associated with insulin resistance and obesity: 3-hydroxyisobutyrate, branched-chain amino acids, branched-chain ketoacids, and direct catabolic products decreased compared to baseline and placebo [89]; of note, these changes were significantly larger with Tirzepatide than with Dulaglutide, and they were directly proportional to reductions in HbA1c, to indices of insulin resistance, and to proinsulin levels [90].

The beneficial effects of Tirzepatide are associated with a good safety profile. In the SURPASS 1–5 and SURMOUNT-1 clinical trials, the main adverse events occurring with Tirzepatide were mild-to-moderate gastrointestinal disorders during the initial dose-escalation phase without clinically significance evidence or severe hypoglycemia (<54 mg/dL; <3 mmol/L). Specifically, nausea, diarrhea, and vomiting occurred. The safety profile was overall consistent with that of GLP-1 receptor agonists (Table 7).

The beneficial influence of Tirzepatide on cardiometabolic parameters suggests the availability, in the near future, of a new weapon effective against cardiovascular and metabolic disorders [88,91,92,93]. Different phase I, II, and III trials are now ongoing to test the effectiveness and safety of Tirzepatide for its potentials uses (Table 8).

## 7. Conclusions

Tirzepatide is the first-in-class twincretin, a peptide with agonist action on both GLP-1 and GIP receptors. The randomized phase III trials SURPASS 1–5 and SURPASS J-mono demonstrated its effectiveness and safety in once-weekly administration in patients with T2DM compared with placebo and other hypoglycemic drugs. In particular, in SURPASS-4, where it was compared to insulin glargine in patients with T2DM, Tirzepatide reduced HbA1c up to −2.58% from baseline values at the 15 mg dose. In the SURPASS-1 trial, up to 52% of participants affected by T2DM achieved normoglycemic status (HbA1c < 5.7%; <39 mmol/mol). Its safety profile was consistent with that of GLP-1 receptor agonists without significant episodes of hypoglycemia (<54 mg/dL; <3 mmol/L).

Tirzepatide was demonstrated to reduce body weight in a dose-dependent way. In SURMOUNT-1, a randomized phase III trial where Tirzepatide effectiveness was tested in patients with overweight and obesity without T2DM, the percentages of participants who had a significant weight reduction of 5% or more were 85%, 89%, and 91% with Tirzepatide 5 mg, 10 mg, and 15 mg, respectively, and a robust weight loss of −20.9% was achieved with Tirzepatide 15 mg [94]. In all studies, reductions in BMI and waist circumference were demonstrated, with improvements in BP values, LDL, and triglycerides. In particular, the SURMOUNT-1 trial revealed a mean reduction in waist circumference of 18.5 cm with Tirzepatide 15 mg; in the SURPASS-5 trial, the mean change in SBP was −6.1 to −12.6 mmHg in patients treated with Tirzepatide.

On 13 May 2022, the Food and Drug Administration (FDA) approved Tirzepatide once-a-week subcutaneous injections (with the dose adjusted based on tolerability to achieve blood glucose targets) as monotherapy or combination therapy, with diet and physical exercise, to achieve better glycemic blood levels in patients affected by T2DM [95,96]. In trials comparing Tirzepatide to other diabetes drugs, patients who received Tirzepatide 15 mg had a 0.5% greater reduction in HbA1c than those who received Semaglutide, a 0.9% greater reduction than those who received insulin degludec, and a 1.0% greater reduction than those who received insulin glargine. It is not fully clear whether the other doses are useful for physicians in clinical practice. The adverse events associated with Tirzepatide include nausea, vomiting, diarrhea, constipation, upper abdominal discomfort, decreased appetite, and abdominal pain; these effects are often observed in irritable bowel syndrome and could suggest a potential action of Tirzepatide on the gut microbiota.

The encouraging results of the SURMOUNT-1 trial and the ongoing trials on Tirzepatide are promising for its future application as a medication for obesity, heart failure, and NAFLD. In addition to drugs that act as dual agonists, such as Tirzepatide, preclinical studies have demonstrated that triagonists, achieving a concurrent activation of GLP-1, GIP, and glucagon receptors, normalize body weight in obese mice and enhance energy expenditure in a manner superior to that of monoagonists of the GLP-1 receptor and dual agonists acting on GLP-1 and GIP receptors [97]. Therefore, the unimolecular triple agonism [98,99,100] could soon represent a new standard for pharmaceutical interventions.

## Figures and Tables

**Figure 1 ijms-23-14631-f001:**
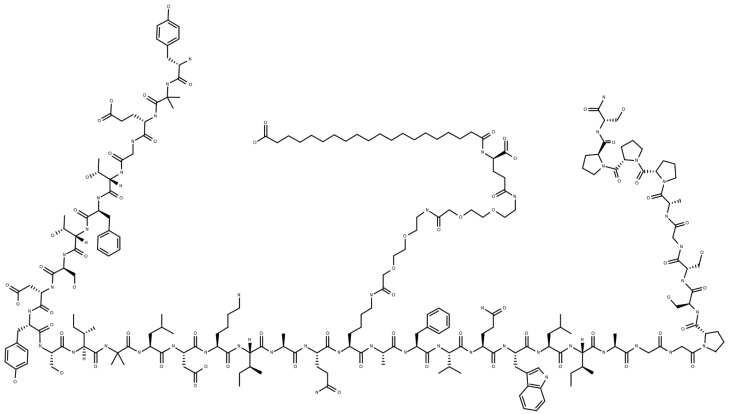
Structure of Tirzepatide (39 amino acids), a dual GIP/GLP-1 receptor agonist (twincretin) that has two non-coded amino acid residues (α-aminoisobutyric acid) at positions 2 and 13 and is acylated on K20 with a γGlu-2×OEG linker and C18 fatty diacid moiety.

**Figure 2 ijms-23-14631-f002:**
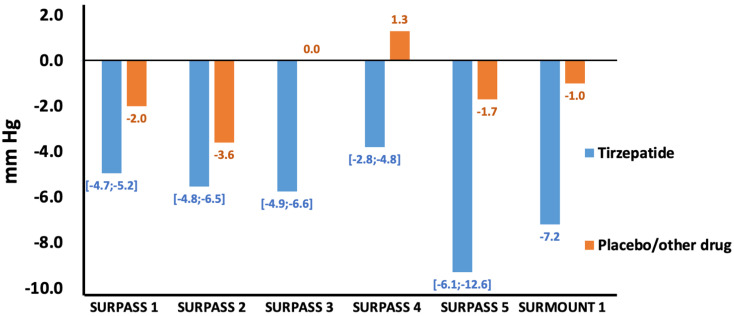
Mean variations in systolic blood pressure (SBP) with Tirzepatide vs. placebo/other drugs in the SURPASS-1 [38], SURPASS-2 [41], SURPASS-3 [44], SURPASS-4 [52], SURPASS-5 [56], and SURMOUNT-1 [66] clinical trials.

**Table 1 ijms-23-14631-t001:** Results of the SURPASS-1 trial at 40 weeks [26]. Efficacy analyses were performed in the modified intention-to-treat (mITT) population, which includes all randomized participants receiving at least one treatment dose; * *p* < 0.0001; ** *p* = 0.72; *** *p* = 0.22.

Outcomes		Tirzepatide 5 mg (n = 121)	Tirzepatide 10 mg (n = 121)	Tirzepatide 15 mg (n = 120)	Placebo (n = 113)
HbA1c (%)	Baseline	7.97	7.88	7.88	8.08
	From baseline	−1.87 *	−1.89 *	−2.07 *	0.04 **
	Versus placebo	−1.91 *	−1.93 *	−2.11 *	-
Weight (Kg)	From baseline	−7.0 *	−7.8 *	−9.5 *	−0.7 ***
	Versus placebo	−6.3 *	−7.1 *	−8.8 *	-

**Table 2 ijms-23-14631-t002:** Results of the SURPASS-2 trial at 40 weeks [41]. Efficacy analyses were performed in the mITT population; * *p* < 0.001; ** *p* = 0.02.

Outcomes		Tirzepatide 5 mg (n = 470)	Tirzepatide 10 mg (n = 469)	Tirzepatide 15 mg (n = 470)	Semaglutide (n = 469)
HbA1c (%)	Baseline	8.32	8.30	8.26	8.25
	From baseline	−2.01	−2.24	−2.30	−1.86
	Versus Semaglutide	−0.15 **	−0.39 *	−0.45 *	-
Weight (Kg)	From baseline	−7.6	−9.3	−11.2	−5.7
	Versus Semaglutide	−1.9 *	−3.6 *	−5.5 *	-

**Table 3 ijms-23-14631-t003:** Results of the SURPASS-3 trial at 52 weeks [44]. Efficacy analyses were performed in the mITT population; * *p* < 0.0001.

Outcomes		Tirzepatide 5 mg (n = 358)	Tirzepatide 10 mg (n = 360)	Tirzepatide 15 mg (n = 358)	Insulin Degludec (n = 359)
HbA1c (%)	Baseline	8.17	8.19	8.21	8.13
	From baseline	−1.93	−2.20	−2.37	−1.34
	Versus insulin degludec	−0.59 *	−0.86 *	−1.04 *	-
Weight (Kg)	From baseline	−7.5	−10.7	−12.9	2.3
	Versus insulin degludec	−9.8 *	−13 *	−15.2 *	-

**Table 4 ijms-23-14631-t004:** Results of the SURPASS-4 trial at 52 weeks [52]. Efficacy analyses were performed in the mITT population; * *p* < 0.0001.

Outcomes		Tirzepatide 5 mg (n = 326)	Tirzepatide 10 mg (n = 321)	Tirzepatide 15 mg (n = 334)	Insulin Glargine (n = 978)
HbA1c (%)	Baseline	8.52	8.60	8.52	8.51
	From baseline	−2.24	−2.43	−2.58	−1.44
	Versus insulin glargine	−0.80 *	−0.99 *	−1.14 *	
Weight (Kg)	From baseline	−7.1	−9.5	−11.7	1.9
	Versus insulin glargine	−9.0 *	−11.4 *	−13.5 *	

**Table 5 ijms-23-14631-t005:** Results of the SURPASS-5 at 40 weeks [56]. Efficacy analyses were performed in the mITT population; * *p* < 0.001.

Outcomes		Tirzepatide 5 mg (n= 116)	Tirzepatide 10 mg (n= 119)	Tirzepatide 15 mg (n= 120)	Placebo (n= 120)
HbA1c (%)	Baseline	8.30	8.36	8.22	8.38
	From baseline	−2.11	−2.40	−2.34	−0.86
	Versus placebo	−1.24 *	−1.53 *	−1.47 *	
Weight (Kg)	From baseline	−5.4	−7.5	−8.8	1.6
	Versus placebo	−7.1 *	−9.1 *	−10.5 *	

**Table 6 ijms-23-14631-t006:** Results of the SURMOUNT-1 trial at 72 weeks [66]. Efficacy analyses were done in the mITT population; * *p* < 0.001.

Outcomes		Tirzepatide 5 mg (n = 630)	Tirzepatide 10 mg (n = 636)	Tirzepatide 15 mg (n = 630)	Placebo (n = 643)
Body Weight (Kg)	Baseline	102.9	105.8	105.6	104.8
	From baseline (%)	−15	−19.5	−20.9	−3.1
	Versus placebo (%)	−11.9 *	−16.4 *	−17.8 *	
Waist circumference (cm)	From baseline	−14	−17.7	−18.5	−4
	Versus placebo	−10.1 *	−13.8 *	−14.5 *	

**Table 7 ijms-23-14631-t007:** Most commonly reported adverse events of Tirzepatide in clinical trials.

	Nausea (%)	Diarrhea (%)	Vomiting (%)
SURPASS-1	12–18	12–14	2–6
SURPASS-2	17–22	13–16	6–10
SURPASS-3	12–24	15–17	6–10
SURPASS-4	12–23	13–22	5–9
SURPASS-5	13–18	12–21	7–13
SURMOUNT-1	25–33	19–23	8–12

**Table 8 ijms-23-14631-t008:** Ongoing clinical trials testing Tirzepatide.

**SYNERGY-NASH** **(phase II)**	A Randomized, Double-Blind, Placebo-Controlled Phase 2 Study Comparing the Efficacy and Safety of Tirzepatide Versus Placebo in Patients with Nonalcoholic Steatohepatitis (NASH).
**SURPASS-CVOT** **(phase III)**	The Effect of Tirzepatide Versus Dulaglutide on Major Adverse Cardiovascular Events in Patients with Type 2 Diabetes.
**SURPASS-PEDS** **(phase III)**	Efficacy, Safety, and Pharmacokinetics/Pharmacodynamics of Tirzepatide in Pediatric and Adolescent Participants with Type 2 Diabetes Mellitus Inadequately Controlled with Metformin, or Basal Insulin, or Both.
**SURPASS-6** **(phase III)**	The Effect of the Addition of Tirzepatide Once Weekly Versus Insulin Lispro Three Times Daily in Participants with Type 2 Diabetes Inadequately Controlled on Insulin Glargine (U100) with or without Metformin.
**SURMOUNT-2** **(phase III)**	Efficacy and Safety of Tirzepatide Once Weekly in Participants with Type 2 Diabetes Who Have Obesity or Are Overweight.
**SURMOUNT-3** **(phase III)**	Efficacy and Safety of Tirzepatide Once Weekly Versus Placebo After an Intensive Lifestyle Program in Participants without Type 2 Diabetes Who Have obesity or Are Overweight with Weight-Related Comorbidities.
**SURMOUNT-4** **(phase III)**	Efficacy and Safety of Tirzepatide Once Weekly Versus Placebo for Maintenance of Weight Loss in Participants Without Type 2 Diabetes Who Have Obesity or Are Overweight with Weight-Related Comorbidities: A Randomized, Double-Blind, Placebo-Controlled Trial.
**SURMOUNT-CN** **(phase III)**	Efficacy and Safety of Tirzepatide Once Weekly in Chinese Participants Without Type 2 Diabetes Who Have Obesity or Are Overweight with Weight-Related Comorbidities: A Randomized, Double-Blind, Placebo-Controlled Trial.
**SURMOUNT-J** **(phase III)**	Efficacy and Safety of Once-Weekly Tirzepatide in Participants with Obesity Disease.
**SUMMIT** **(phase III)**	Study Comparing the Efficacy and Safety of Tirzepatide Versus Placebo in Patients with Heart Failure with Preserved Ejection Fraction and Obesity.
**SURPASS-EARLY** **(phase IV)**	Study to Evaluate the Long-Term Efficacy and Safety of Tirzepatide Compared with Intensified Conventional Care in Adults When Initiating Treatment Early in the Course of Type 2 Diabetes.

## Data Availability

Not applicable.

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
