# Peer review of "Tirzepatide: A Systematic Update"

_ijms, 2022, doi:10.3390/ijms232314631_

Round 1

Reviewer 1 Report

Summary: The authors did good job summarizing the results of the main clinical trials investigating Tirzepatide, and emphasizing novel insights important for clinical practice.   They reviewed SURPASS 1-5 and SURMOUNT 1 trial and focused on the primary outcomes of Hb1C and weight.  This is an important review as this medication is getting lots of attention due to the impressive results from the different clinical trials, comparing to other standard of care medications.

General concept: Overall, well written review, fairly complete review.  The authors highlighted the main primary outcomes from each trial and presented in charts which is easy to follow.  The authors also reported interesting secondary outcomes from some of the reviews that are clinically relevant, and reported about effect on hypertension in a combined graph in figure 2 which was easy to follow and well visualized.  I also liked the combined chart with most common side effects reported. 

Specific comments: none

Author Response

Thanks!

I am also reporting below the responses to the Editor's comments:

  1. This Review is insufficient developed.

R: We thank the Editor for these comments. Please note that the number of available studies concerning Tirzepatide is very limited as it is a brand-new topic (Tirzepatide has been approved by the FDA only in May 2022).

  1. Introduction is too poor (not presenting the main aspects implied by the topic), aim of the study is not well defined as no real/complete analysis was done throughout the manuscript, no novelty/special aspects are described.

R: We have expanded the manuscript in all its sections (100 references), including the Introduction, as suggested. The aim of the study was to make a review on Tirzepatide evaluating results of SURPASS 1, SURPASS 2, SURPASS 3, SURPASS 4, SURPASS 5, SURPASS J-Mono and SURMOUNT 1 trial, which are the main 7 clinical trials that are available at this moment for Tirzepatide. We respectfully believe that being this a review and not an original article, we cannot refer to novelty in the Introduction.

  1. Instructions for Authors were not read/respected in writing the manuscript (table settings).

R: Table settings have been rectified; in the revised version, we used the template provided by the Journal.

  1. Figure 1 is blurred, a specific program for writing structural formula would be helpful. Prints screen it, not save it in any format, as you will lose clarity of the image.

R: We now uploaded a 1200 pixel/inches figure, which will be published in the final version of the published paper. The blurring could be an optical effect given by different types of bonds represented in the figure.

  1. First 5 tables are presenting results of a single study (where is the Review of the literature?), no references inserted for those tables. Empty cells in tables are not allowed for a scientific paper. These tables could have been merged because they describe the same type of parameters. Presenting the results of some studies does not mean an analysis. There are no real , complete comparisons between them.

R: We respectfully want to point out that the First 5 tables are evidently referring to 5 different studies that evaluate 5 different therapeutic conditions (also, please note that this is not a meta-analysis, but a review, so we are not supposed to analyze or compare these studies):

SURPASS 1: tirzepatide versus placebo in patients affected by type 2 diabete mellitus

SURPASS 2: tirzepatide versus semaglutide in patients affected by type 2 diabete mellitus

SURPASS 3: tirzepatide versus once-daily insulin degludec as add-on to metformin with or without SGLT2 inhibitors

SURPASS 4: Tirzepatide versus insulin glargine in patients affected by type 2 diabetes mellitus

SURPASS 5: Tirzepatide vs Placebo Added to Titrated Insulin Glargine on Glycemic Control in Patients with Type 2 Diabetes

Reference to every single study has been added.

  1. Figure 2 is a blurred graphic, presenting results of a specific research. Where is the Review?, in this case. Or this is the entire analysis the authors aimed for providing? Figure 2 is a focus of one of the secondary endpoints analysed by each SURPASS. Is a comparison between them. Figure can be improved and references can be add.

R: We have provided a revised high-quality version of Figure 2, which is a representation of the effect of Tirzepatide on blood pressure in different trials. References have been added, as requested.

  1. Original graphical part is missing.

R: We respectfully want to point out that the aim of this review is to present in one single manuscript all the data generated in the clinical trials investigating Tirzepatide.

  1. Table 7. As % is presented in ALL columns (2 to 4), why you have not inserted % in the head of the Table, instead of writing this symbol 18 times?

R: Rectified, thanks.

  1. Table 8 is not referenced at all, even it presents data of multiple studies.

R: As clearly stated in the manuscript, Table 8 is recapitulation of the currently ongoing trials on Tirzepatide; being ongoing, these trials have not been published and references to published studies cannot be added.

  1. Conclusions section present again data from literature, no pertinent, focused conclusion of an analysis.

R: We have improved the discussion of the results throughout the manuscript and expanded conclusion and perspectives.

  1. The real manuscript has 10 pages, written in the MDPI format (where the text is occupying 2/3 of a page!).

R: The revised manuscript has 21 pages. Respectfully, we do not understand how a scientific review on a novel topic can even be evaluated by the format used or by the number of pages.

  1. 36 References cannot describe this relevant topic, a much more careful analysis/research of the literature must be done.

R: The revised manuscript has 100 references. As mentioned above, the number of studies concerning tirzepatide is not numerous as it is a brand new topic (Tirzepatide has been approved by the FDA only in May 2022). Nevertheless, we have now added more references in the revised version of the manuscript.

Reviewer 2 Report

Summary of review:

The purpose of "Tirzepatide: an update" by Forzano et al. was to review past and future clinical studies with the twincretin Tirzepadine as a Glucagon-like peptide-1 (GLP-1) and glucose-dependent insulinotropic polypeptide (GIP) agonist. This review presented original data obtained from the five published clinical and intervention studies: SURPASS 1-5 and SURMONT-1. A perspective is presented on future clinical interventions. Two tables discuss systolic blood pressure and 3 side effects for these 6 studies.

This document reflects the first approval obtained from the FDA on May 13, 2022 or the NDA (New drug application) reviewed by the FDA committees. All pharmacological evaluation is supported by Eli Lilly. 

The major weaknesses and/or deficiencies are as follows: 

Reading the excerpt (below) from the FDA approval, "Mounjaro is a first-in-class drug that activates both GLP-1 and GIP receptors, leading to improved blood glucose control. Mounjaro is administered by injection under the skin once a week, with the dose adjusted based on tolerability to achieve blood glucose targets

On average, patients randomized to receive the maximum recommended dose of Mounjaro (15 milligrams) had a 1.6 percent greater reduction in hemoglobin A1c (HbA1c) (a measure of blood glucose control) than placebo when used as a stand-alone therapy, and a 1.5 percent greater reduction than placebo when used in combination with long-acting insulin. In trials comparing Mounjaro to other diabetes drugs, patients who received the maximum recommended dose of Mounjaro had a 0.5% greater reduction in HbA1c than semaglutide, a 0.9% greater reduction than insulin degludec and a 1.0% greater reduction than insulin glargine. "

This fact or quote should be discussed more before presenting only the conclusion (at least the bolded sentences and the active dose of 15 mg). An higher dosage is better or not  safe ?

-Is the efficacy of 5 and 10 mg useful or not for the physician?  

-Meta-analyses are published now, they should be presented and discussed. Since meta-analyses are better than RCTs in terms of evidence. 

-Results after 2 years of treatment have been published. Surmont-1 cited by the authors is 72 weeks. The duration of efficacy is important for this type of drug in terms of control of diabetes over time but also in terms of anorexigenic effect which was the failure (heart valves, Papers by Frachon I et al.) of a previous drug like mediator in France with the same "weight loss" application. 

-The adverse effects are mainly gastrointestinal. But the FDA has stated that decreased appetite, constipation, upper abdominal discomfort and abdominal pain are present. All of these effects suggest an effect on the gut microbiota and possible intestinal bowel syndrome ?

-For the first 6 tables, statistical significance as it appears should be presented. Severity and discontinuation of treatment are not discussed. Are these results obtained in ITT or PP?

In the Surpass-4 study by Del prado et al. tirzepatide, a dual agonist of GIP and GLP-1 receptors, significantly improved glucose control, reduced body weight, and improved the cardiovascular risk profile. But this study has some limitations, as they said. First, the interventions were not blinded because of differences in devices and dose escalation schemes. Second, because the variable treatment period beyond 52 weeks was designed to collect longer-term safety data and reach a predefined number of MACE-4s, not all participants were treated for 104 weeks. These data are not reported in some meta-analyses. Dose escalation is required for phase II and not phase 3, could you present the classification for discussion.

BMI is presented in the baseline demographics table. It is used as an inclusion criterion. This is an important point that may vary in future studies, such as application for pre-diabetes in obese patients, a discussion on a lower BMI should be made.

Tirzepatide and cardiovascular event risk assessment: a recently published pre-specified meta-analysis in 2022 by Sattar N. et al. The pre-specified primary objective of this meta-analysis was the comparison of time to first occurrence of confirmed four-component major adverse cardiovascular events (MACE-4; cardiovascular death, myocardial infarction, stroke, and unstable hospitalized angina) between pooled tirzepatide and control groups.

The exclusion of people with unstable cardiovascular disease (class IV heart failure) is a limitation, as they were excluded from the first 6 studies and many drugs lost their therapeutic claims (only type 2 diabetes today). The review of the obesity application is ongoing, as it is submitted by the FDA. 

My opinion is that only death is an outcome for the future and that the MACE-4 used in this study is just a composite endpoint that is not appropriate.

In this study, the hazard ratio (subgroup analyses) for other countries is 0.9 (O.61-1.31) and is very different from the US results. What other studies are there for worldwide application and why?

In the article not cited by this manuscript, Sattar, N. et al. Cardiovascular, mortality, and kidney outcomes with GLP-1 receptor agonists in patients with type 2 diabetes: a systematic review and meta-analysis of randomised trials. Lancet Diabetes Endocrinol. 9, 653-662 (2021). It was reported: "Despite the favorable effects of tirzepatide on a range of cardiovascular risk factors, to date, its cardiovascular safety has been reported in only one trial, SURPASS-4 (ref. 11). This trial, which compared tirzepatide therapy with T2DM at increased cardiovascular risk, suggested no significant difference in the incidence of major cardiovascular events. 

In conclusion, this review should present a critical update with the negatives as I have discussed here and articles all with a conflict of interest such as the Lilly publications.

Author Response

Summary of review:

The purpose of "Tirzepatide: an update" by Forzano et al. was to review past and future clinical studies with the twincretin Tirzepadine as a Glucagon-like peptide-1 (GLP-1) and glucose-dependent insulinotropic polypeptide (GIP) agonist. This review presented original data obtained from the five published clinical and intervention studies: SURPASS 1-5 and SURMONT-1. A perspective is presented on future clinical interventions. Two tables discuss systolic blood pressure and 3 side effects for these 6 studies.

This document reflects the first approval obtained from the FDA on May 13, 2022 or the NDA (New drug application) reviewed by the FDA committees. All pharmacological evaluation is supported by Eli Lilly. 

The major weaknesses and/or deficiencies are as follows: 

Reading the excerpt (below) from the FDA approval, "Mounjaro is a first-in-class drug that activates both GLP-1 and GIP receptors, leading to improved blood glucose control. Mounjaro is administered by injection under the skin once a week, with the dose adjusted based on tolerability to achieve blood glucose targets. On average, patients randomized to receive the maximum recommended dose of Mounjaro (15 milligrams) had a 1.6 percent greater reduction in hemoglobin A1c (HbA1c) (a measure of blood glucose control) than placebo when used as a stand-alone therapy, and a 1.5 percent greater reduction than placebo when used in combination with long-acting insulin. In trials comparing Mounjaro to other diabetes drugs, patients who received the maximum recommended dose of Mounjaro had a 0.5% greater reduction in HbA1c than semaglutide, a 0.9% greater reduction than insulin degludec and a 1.0% greater reduction than insulin glargine. "

This fact or quote should be discussed more before presenting only the conclusion (at least the bolded sentences and the active dose of 15 mg). An higher dosage is better or not  safe ? -Is the efficacy of 5 and 10 mg useful or not for the physician?  

 R: We fully agree with this Reviewer. We now discuss these points in the revised version of the manuscript.

-Meta-analyses are published now, they should be presented and discussed. Since meta-analyses are better than RCTs in terms of evidence. 

R: We thank this Reviewer for her/his pertinent remark. We now also include meta-analyses in the revised version of the manuscript.

-Results after 2 years of treatment have been published. Surmont-1 cited by the authors is 72 weeks. The duration of efficacy is important for this type of drug in terms of control of diabetes over time but also in terms of anorexigenic effect which was the failure (heart valves, Papers by Frachon I et al.) of a previous drug like mediator in France with the same "weight loss" application. 

R: We thank this Reviewer for this interesting note. We now also mention this paper by Frachon et al. in the SURMOUNT-1 paragraph.

-The adverse effects are mainly gastrointestinal. But the FDA has stated that decreased appetite, constipation, upper abdominal discomfort and abdominal pain are present. All of these effects suggest an effect on the gut microbiota and possible intestinal bowel syndrome ?

R: We thank this Reviewer for this suggestion. In the revised version of the manuscript we discuss this important point.

-For the first 6 tables, statistical significance as it appears should be presented. Severity and discontinuation of treatment are not discussed. Are these results obtained in ITT or PP?

R: We thank this Reviewer for this comment . In the revised version of the manuscript we update the first 6 tables, as requested.

In the Surpass-4 study by Del prado et al. tirzepatide, a dual agonist of GIP and GLP-1 receptors, significantly improved glucose control, reduced body weight, and improved the cardiovascular risk profile. But this study has some limitations, as they said. First, the interventions were not blinded because of differences in devices and dose escalation schemes. Second, because the variable treatment period beyond 52 weeks was designed to collect longer-term safety data and reach a predefined number of MACE-4s, not all participants were treated for 104 weeks. These data are not reported in some meta-analyses. Dose escalation is required for phase II and not phase 3, could you present the classification for discussion. BMI is presented in the baseline demographics table. It is used as an inclusion criterion. This is an important point that may vary in future studies, such as application for pre-diabetes in obese patients, a discussion on a lower BMI should be made.

R: We thank this Reviewer for another helpful and pertinent comment. We now better discuss the limitations of the SURPASS-4 trial by Del Prato et al.

Tirzepatide and cardiovascular event risk assessment: a recently published pre-specified meta-analysis in 2022 by Sattar N. et al. The pre-specified primary objective of this meta-analysis was the comparison of time to first occurrence of confirmed four-component major adverse cardiovascular events (MACE-4; cardiovascular death, myocardial infarction, stroke, and unstable hospitalized angina) between pooled tirzepatide and control groups.

The exclusion of people with unstable cardiovascular disease (class IV heart failure) is a limitation, as they were excluded from the first 6 studies and many drugs lost their therapeutic claims (only type 2 diabetes today). The review of the obesity application is ongoing, as it is submitted by the FDA. 

My opinion is that only death is an outcome for the future and that the MACE-4 used in this study is just a composite endpoint that is not appropriate.

In this study, the hazard ratio (subgroup analyses) for other countries is 0.9 (O.61-1.31) and is very different from the US results. What other studies are there for worldwide application and why?

In the article not cited by this manuscript, Sattar, N. et al. Cardiovascular, mortality, and kidney outcomes with GLP-1 receptor agonists in patients with type 2 diabetes: a systematic review and meta-analysis of randomised trials. Lancet Diabetes Endocrinol. 9, 653-662 (2021). It was reported: "Despite the favorable effects of tirzepatide on a range of cardiovascular risk factors, to date, its cardiovascular safety has been reported in only one trial, SURPASS-4 (ref. 11). This trial, which compared tirzepatide therapy with T2DM at increased cardiovascular risk, suggested no significant difference in the incidence of major cardiovascular events. 

R: We thank this Reviewer for her/his insightful suggestions. We now better discuss these aspects and we include also two trials conducted in Japan. We want to specify that as noted in the meta-analysis by Sattar et al (Nat Med 2022, included in our review) "Despite the favorable effects of tirzepatide on a range of cardiovascular risk factors, to date, its cardiovascular safety has been reported in only one trial, SURPASS-4. This trial, which compared tirzepatide therapy to insulin glargine 100U ml−1 treatment in people with T2DM at increased cardiovascular risk, suggested no significant difference in the incidence of major cardiovascular events". Concerning subgroup analysis, in the same manuscript by Sattar et al. the Authors specified that “Subgroup analyses for the primary outcome of MACE-4 by sex, age, baseline HbA1c, race, US or non-US clinical sites and baseline sodium glucose co-transporter 2 inhibitor (SGLT-2i) use demonstrated no significant effect modification (all Pinteraction>0.1)”.

In conclusion, this review should present a critical update with the negatives as I have discussed here and articles all with a conflict of interest such as the Lilly publications.

R: We thank this Reviewer for the time spent reading our paper. In the revised version, we have included several critical aspects of Tirzepatide, as requested.

Round 2

Reviewer 2 Report

I appreciate the answers done by the authors. Thanks